# Hypoxia as a Double-Edged Sword to Combat Obesity and Comorbidities

**DOI:** 10.3390/cells11233735

**Published:** 2022-11-23

**Authors:** Ruwen Wang, Qin Sun, Xianmin Wu, Yiyin Zhang, Xiaorui Xing, Kaiqing Lin, Yue Feng, Mingqi Wang, Yibing Wang, Ru Wang

**Affiliations:** School of Exercise and Health, Shanghai Frontiers Science Research Base of Exercise and Metabolic Health, Shanghai University of Sport, Shanghai 200438, China

**Keywords:** adipose tissue, hypoxic, obesity, PO_2_, O_2_

## Abstract

The global epidemic of obesity is tightly associated with numerous comorbidities, such as type II diabetes, cardiovascular diseases and the metabolic syndrome. Among the key features of obesity, some studies have suggested the abnormal expansion of adipose-tissue-induced local endogenous hypoxic, while other studies indicated endogenous hyperoxia as the opposite trend. Endogenous hypoxic aggravates dysfunction in adipose tissue and stimulates secretion of inflammatory molecules, which contribute to obesity. In contrast, hypoxic exposure combined with training effectively generate exogenous hypoxic to reduce body weight and downregulate metabolic risks. The (patho)physiological effects in adipose tissue are distinct from those of endogenous hypoxic. We critically assess the latest advances on the molecular mediators of endogenous hypoxic that regulate the dysfunction in adipose tissue. Subsequently we propose potential therapeutic targets in adipose tissues and the small molecules that may reverse the detrimental effect of local endogenous hypoxic. More importantly, we discuss alterations of metabolic pathways in adipose tissue and the metabolic benefits brought by hypoxic exercise. In terms of therapeutic intervention, numerous approaches have been developed to treat obesity, nevertheless durability and safety remain the major concern. Thus, a combination of the therapies that suppress endogenous hypoxic with exercise plans that augment exogenous hypoxic may accelerate the development of more effective and durable medications to treat obesity and comorbidities.

## 1. Introduction

### 1.1. The Potential Connection between Endogenous Hypoxic or Hyperoxia in Adipose Tissues and Obesity

Over the past half century, the global prevalence of obesity has increased significantly and obesity has become a global pandemic. According to the report on Nutrition and Chronic Diseases (NCDS) in China, the percentage of overweight and obesity in all age groups continued to rise, and more than half of the adults were either overweight or obese. Notably, mortality caused by overweight and obesity accounted for 11.1% of all deaths related to NCDS [1]. Obesity is featured by pathological increment in the number and volume of adipocytes, with dysregulated ratio between adipose tissue and other tissues [2]. It is caused by multiple factors, e.g., genetic variations, excessive lipogenesis and subsequently abnormal lipid accumulation in adipose tissue. Obesity is frequently accompanied with numerous metabolic comorbidities, such as non-alcoholic fatty liver disease (NAFLD), diabetes mellitus and hypertension, and is tightly associated with an increased risk of certain types of cancer [3,4]. The soaring prevalence of obesity has significant propelled investigation into the biological functions of adipose tissue, the pathogenesis of obesity and development of the specific therapeutics.

Adipose tissue plays pivotal physiological functions in vivo, serving as the source of energy storage and as a buffer against external shocks [5]. Furthermore, as one of the major endocrine organs, adipose tissues secrete a variety of factors into circulation and they mediate crosstalk between adipose tissue and other organs to maintain homeostasis in vivo [6,7]. The expression and secretion of these adipokines are dramatically altered during the progression of obesity, while dysfunction of white adipose tissue contributes to the development of obesity-related diseases [8].

As well as the dysregulated signaling network of adipokines, the state of hypoxic or hyperoxia under different circumstances in the progression of obesity is an important feature. Some studies suggest that in genetic (*ob/ob* mice) and diet-induced obesity models, the adipose tissues are actually in a state of hypoxic, as reflected by an increase in the fluorescence emitted by probes or endogenous markers that respond to hypoxic [9,10]. Lower oxygen partial pressure in adipose tissue was quantified in obese mice, along with the expansion in the tissue mass [9,11,12,13]. Thus, some studies proposed that the subsequent hypoxic in adipose tissue triggered adipose tissue dysfunction [14], which is termed “endogenous hypoxic”.

Nevertheless, the concept of adipose tissue hypoxic in obesity is still not firmly established. One study has suggested that diet-induced weight loss significantly lowered abdominal subcutaneous adipose tissue oxygen tension (pO2) in fifteen overweight and obese individuals [15]. Thus, the findings in humans require further investigation as the development of obesity in human and rodents may involve distinct mechanisms. Alternatively, the genetic *ob/ob* mouse and diet-induced obesity model may not fully recapitulate the heterogeneity of the etiology of obesity in human.

Induction of hypoxic activates the expression of hypoxic-inducible factors (HIFs) and they stimulate the transcription of the downstream genes [16], including the platelet-derived growth factor (PDGF) [17], Leukotriene B4 (LTB4) [18] and macrophage inflammation factor 1 (MIF1) [19]. Activation of these factors rewires the physiological functions and signaling pathways in adipose tissues, including angiogenesis, insulin sensitivity, immune responses and lipid homeostasis [20]. The altered factors and signaling pathways are responsible for the impaired angiogenesis, inflammation, fibrosis and metabolic inflexibility [21,22,23], and it eventually gives rise to the onset of metabolic diseases [23].

### 1.2. Hypoxic Training Ameliorates the Symptoms in Obesity

In addition to the locally endogenous hypoxic and hyperoxia in adipose tissue induced by obesity, hypoxic trainings generate exogenous hypoxic and subsequently induce metabolic benefits to human body [24]. A myriad of exercise interventions has effectively promoted weight loss, systemically normalized metabolic homeostasis and resolved inflammation [25]. Exercise enhances cardiopulmonary function and reduces the risk of cardiovascular disease. In addition, it is the most effective way to control weight and calorie intake as well [26]. Among the various forms of training, plateau training program began in the mid-1950s with the athletes competing at altitudes above 1500 m [27]. The results indicated that the performances of athletes who participated in plateau training program were significantly improved compared with those who did not [27].

Due to the limitations in plateau training (geographic location, high-altitude sickness, etc.), researchers have renovated the protocols of plateau training and proposed new hypoxic training models and concepts that de-coupled the altitude of living from the training of the athletes, such as “living high and training low”, and “living low and training high”, etc. [28]. Hypoxic exercise has reduced body weight, increased vascular blood flow rate [29], improved metabolic syndrome, etc. [30,31,32]. Meanwhile, the combination of hypoxic and exercise can shorten the training time, while achieving weight loss in a more profound way [33].

### 1.3. Revisit of the Previous Studies and Therapeutic Interventions

Understanding the mechanism of endogenous and exogenous hypoxic in obesity facilitates the investigators to harness the properties to design novel therapies. Nevertheless, most of the previous studies have focused on the effect of hypoxic or hypoxic training in improving the metabolic function. The mechanisms, especially the effect of oxygen on metabolic homeostasis, remain partially understood.

In addition, some seminal studies have generated inconsistent results on the effect of hypoxic in obesity. For instance, multiple studies have demonstrated that the oxygen partial pressure in adipose tissues of people with obesity was lower than that of lean subjects [34]. The higher expression of hypoxic-regulated genes was associated with insulin resistance [35,36]. Nevertheless, Goossens et al. have pointed out that the oxygen partial pressure in adipose tissues of people with obesity was actually higher than that of lean subjects [37], while diet-induced weight loss decreases oxygen partial pressure in adipose tissues [15]. The inconsistency may be explained by lower adipose tissue oxygen consumption in people with obesity, which underlies the higher adipose tissue oxygen partial pressure [37,38]. Alternatively, the distinct etiology of obesity, such as the “obesity” driven by different master regulators or lifestyles (e.g., deficiency in Leptin receptor) may exhibit different status of oxygen concentration in adipose tissues.

In this review, we will discuss the distinctive mechanisms between endogenous hypoxic (interstitial hypoxic due to excessive expansion of adipose tissue) and exogenous hypoxic exposure from new perspectives. In contrast, an exogenous hypoxic intervention promotes utilization of lipid and reduces adipogenesis. What are the precise oxygen concentrations that are sensed by the cells in vivo?

We will also discuss the pivotal pathological alterations in adipose tissue under hypoxic conditions during the progression of obesity, especially the broad effects of hypoxic on white adipocytes, and its implications for the functioning of adipose tissues. Importantly, we will extensively discuss the optimal methods of hypoxic training for obese patients.

Moreover, in terms of translational perspective, currently there are few approved safe and durable drugs for treatment of obesity. One exception are peptide-based drugs such as Tirzepatide, the agonist against dual GIP/GLP-1 receptor, which significantly lowers body weight in people with obesity in Phase III trials [24]. Tirzepatide systemically normalizes the glucose and lipid metabolism in a multiple tissue way.

As new insights into the therapeutics, we also propose strategies to harness the aforementioned double-faced properties of hypoxic in obesity and comorbidities to combat these diseases. The mediators that are critical for endogenous hypoxic can be targeted and the therapies combined with hypoxic training may effectively induce weight loss and ameliorate comorbidities in people with obesity.

## 2. Endogenous Hypoxic in Adipose Tissue

### 2.1. Types of Adipose Tissue and Their Functions

Adipose tissue is categorized into three main types: white adipose tissue (WAT), brown adipose tissue (BAT) and beige adipose tissue [39]. WAT served as thermal insulation and mechanical cushion, and is mainly composed of white adipocytes, pre-adipocytes, endothelial cells and immune cells, as well as mesenchymal stem cells. [40]. White adipocytes exhibit distinct unicameral morphology, along with few mitochondria. In contrast, lipid droplets occupy most of their intracellular spaces. [22,41,42]. WAT is a highly active endocrine organ and secretes a variety of regulatory peptides and adipokines, such as leptin [43,44], adiponectin [44], chemerin [45], transforming growth factor (TGF)-β2 [46] and tumor necrosis factor-α(TNF-α) [47]. They regulate the structural and functional remodeling of WAT and coordinate with the factors in liver, skeletal muscle, pancreas, etc., to maintain energy homeostasis and metabolic health [48,49].

BAT is differentiated from progenitors expressing Myogenic factor 5 (*Myf5*) and Paired box protein 7 (*Pax7*) genes in the mesoderm [50,51]. They contain multilocular lipid droplets with far more mitochondria than white adipocytes do, and have the central nucleus [42]. Notably, uncoupling protein 1 (UCP1) in mitochondria of BAT de-couples ATP synthesis with oxidative phosphorylation and enhances adaptive thermogenesis [52,53]. BAT secretes multiple batokines, such as fibroblast growth factor 21 [54], interleukin (IL)-6 [55], neuregulin-4 [56], vascular endothelial growth factor B (VEGF-B) [57] and bone-morphogenetic protein(BMP)-8B [58]. They target distant tissues and organs and are critical to maintaining metabolic homeostasis, protection of tissues and development [59], etc. Excision of BAT caused significant weight gain with exacerbated hyperlipidemia, obesity, and increased risk of Type 2 diabetes [60,61] in mice. BAT activity is thus beneficial for metabolic homeostasis and cardiovascular function [62,63].

Beige adipose tissue (most are differentiated from *Myf5^-^* cell lineage [64]) is generated within WAT via: (1) transformation of mature white adipocytes to beige adipocytes, (2) De novo differentiation into new beige adipocytes, and (3) the proliferation of mature beige adipocytes [65,66]. Beige adipocytes also have multilocular droplets and abundant cristae-dense mitochondria, and regulate energy balance by modulating glucose and lipid metabolism [67].

### 2.2. Induction of the Endogenous Hypoxic Environment in Adipose Tissues and the Molecular Mediators

Hypoxic drives the adipose tissue dysfunction and metabolic inflexibility during obesity [13,23,68]. Excessive energy in vivo accelerates de novo adipocyte differentiation (or “adipogenesis”) from precursor cells to meet the requirements in lipid storage [69]. As an adaptive mechanism to increase the buffering capacity of nutrients, lipid droplets in adipocytes cause diminished lipid accumulation in other tissues, thus adipocytes undergo adaptive hypertrophy, and eventually leading to obesity [70]. The perception of excess nutrients by adipose tissue induces the release of epinephrine and norepinephrine from nerve endings in adipose tissue, which bind to β-adrenergic receptors on adipocytes and activate the downstream signaling pathways [71,72]. The activation of catabolic pathways mediated by mitochondrial lipid metabolism increases lipolysis and accelerates thermogenesis, which leads to a surge in oxygen consumption in adipose tissue [73,74,75]. At the same time, the rapid expansion of adipocytes causes insufficient local supply of blood and oxygen, along with increased diffusion distance of O_2_. It causes a gradual decrease in PO_2_ from the bloodstream in adipose tissue to the distal locations (e.g., PO_2_ dropped from 47.9 mmHg to 15.2 mmHg in obese mice) [9,76]. Therefore, local hypoxic microenvironment is generated in adipose tissue when the diameter of adipocyte reaches 100 μm (150–200 μm) [77,78].

Once the endogenous hypoxic environment is generated in adipose tissue, the molecular responses to the hypoxic environment are mediated by HIFs. The HIFs family contains two distinct subunits: α (HIF-1α, HIF-2α and HIF-3α) and β (HIF-1β). HIF-1α is widely expressed in most tissues and organs [79]. Hypoxic inhibits the activity of oxygen-dependent prolyl hydroxylases (PHDs) and thus blocks the hydroxylation of HIF-1α and the subsequent ubiquitin-mediated proteasomal degradation [18,23]. As a result, HIF-1α and HIF-1β form heterodimer, which then binds to the promoter sequences, which are termed hypoxic response elements (HREs) of target genes in the genome [18]. Finally, the heterodimer stimulates transcription of numerous downstream hypoxic-related genes [80,81], including *Socs3* [82], *PDK1* [83], *Sirt2* [84] and *Irak-M* [85]. In the progression of obesity, the downstream cascades cause distorted glucose and lipid metabolism, and influence the development of complications such as NAFLD and Type 2 Diabetes [86].

## 3. Endogenous Hypoxic and Adipose Tissue Dysfunction

At the initial stage of obesity, the body responds to over-nutrition by increasing catabolism and material storage in adipose tissue, so as to maintain homeostasis in the internal environment. Some studies proposed that elevation in oxygen consumption during tissue and adipose tissue expansion is essential to catabolism and therefore elicits acute endogenous hypoxic [80]. Chronically endogenous hypoxic and subsequent adipose tissue dysfunction will develop under the circumstances of long-term high caloric intake [87,88]. Adipose tissue dysfunction is characterized by abnormality in adipocyte proliferation and hypertrophy, inadequate vascularization, immune cell infiltration, ectopic lipid deposition in other tissues and organs (e.g., liver, abdominal visceral depots, skeletal muscle, pancreas), mitochondrial dysfunction and increased production of pro-fibrotic extracellular matrix (ECM) proteins [89,90,91,92]. As shown in Figure 1.

### 3.1. Endogenous Hypoxic Modulates Dysfunction of White Adipose Tissue

#### 3.1.1. Acute Endogenous Hypoxic Induces Adaptive Remodeling of White Adipose Tissue

Acute exposure of hypoxic upregulates the expression of HIF-1α and a series of angiogenic factors, including VEGF-A [93], PDGF and insulin-like growth factor (IGF)-1 [17,94]. Increased angiogenesis in adipose tissue attenuates the negative effects of hypoxic, and supplies more oxygen to adipocytes and eliminates metabolic waste (e.g., the increase in lactate generated by aerobic glycolysis). Meanwhile, acute exposure of hypoxic provokes inflammation responses and also indirectly promotes the proliferation of vascular endothelial cells [95]. For example, obesity induces the secretion of PDGF by inflammatory macrophages in WAT. PDGF stimulates detachment of pericytes (PCs) from blood vessels, which is mediated by PDGFR-β, and thus promotes angiogenesis [96]. Recent studies found that transcription factor 21 (TCF21), the visceral derived adipose stem cells marker, was highly expressed in a visceral pro-inflammatory environment, and promoted the secretion of IL-6 and matrix metalloproteinase (MMP)-dependent ECM remodeling in visceral adipose stem cells [97]. IL-6 induced STAT3 phosphorylation via the JAK family of tyrosine kinases in murine white adipocyte progenitor cells. TGF-β3 expression was inhibited and browning of white adipocytes were stimulated [98]. In fact, increased secretion of IL-6 and MMP-dependent ECM remodeling have been observed in the endogenous hypoxic environment of adipose tissue during obesity [68,99]. TCF21, as the WAT-specific marker gene, may be a potential master regulator in the hypoxic-associated signal network and actively improves WAT function in obesity [97,100].

As a result, acute endogenous hypoxic can induce angiogenesis and remodeling of adipocyte to counteract hypoxic, which is comprised of the multiple layers of negative feedback undertaken in adipose tissue against acute endogenous hypoxic.

#### 3.1.2. Adverse Changes Induced by Chronic Endogenous Hypoxic

##### Aggravation of Inflammation

Chronic expansion of adipose tissue in obesity decreases capillary density [34]. Therefore, reduction in tissue blood perfusion is the prerequisite for chronically endogenous hypoxic, which elicits severe oxidative stress in adipocytes, induces cellular immune response and chronic inflammation [101]. One key mechanism underlying the chronic inflammation is that hypoxic upregulates inflammatory genes in adipose tissue such as *IL-6*, *LEP*, *MCP-1* and *CYR61* through activation of HIF-1α [68,80]. Meanwhile, chronic endogenous hypoxic leads to recruitment of macrophages into WAT, and leukocytes mediate formation of crown-like structures (CLS) [102], which are considered as hallmarks of inflammation [103].

The recent study reported that HFD-induced obesity led to adenine nucleotide translocase (ANT)-mediated uncoupled respiration, which elevated adipocyte oxygen consumption and subsequent transformed WAT into the status of relative hypoxic [80]. Consequently, HIF-1α upregulated chemokines, such as monocyte chemoattractant protein -1 (MCP-1) and LTB4, further aggravating inflammation [80]. Regazzetti et al. showed that hypoxic induced insulin resistance in adipocytes through expression of HIF transcription factors and subsequently inhibited insulin signaling pathway by decreasing phosphorylation of insulin receptor [104]. They proposed that the initial stage of insulin resistance in adipose tissue is caused by hypoxic through inhibition of insulin signaling. Subsequent hypoxic leads to the recruitment of macrophages in adipose tissue and dysregulated expression of adipokines (such as TNF-α), further aggravating insulin resistance in adipose tissue. [104].

Macrophages are generally categorized into two types: classical M1 pro-inflammatory macrophages and M2 anti-inflammatory macrophages [105]. A variety of inflammatory factors secreted by M1 macrophages provoke local inflammatory responses in WAT [106], including leptin, IL-6, TNF-α and IL-1β [68,107]. TNF-α is the first identified adipokine that is secreted by macrophages, and it blocks the effect of insulin in adipocytes [108]. TNF-α phosphorylates Serine^307^ of IRS-1(insulin receptor substrate 1) by activating downstream c-Jun NH2-terminal kinase (JNK), impairing glucose uptake and causing insulin resistance in adipose tissue [109,110,111]. Under hypoxic exposure, white adipocytes potentiate TNF-α-induced upregulates expression of inducible NO synthase and cyclooxygenase-2(COX2), and downregulates peroxisome proliferator-activated receptor γ (PPARγ) and peroxisome proliferator-activated receptor gamma coactivator-1 α (PGC-1α) levels, which is closely associated with inflammation in adipocyte induced by hypoxic under the circumstance of obesity [112].

Recent studies found that the function of M2 macrophages depended on the pathway to stimulate the expression of anti-inflammatory factors and the secretion of the IL-10 in a PPARγ-dependent way (which maintains the insulin sensitivity of adipocytes) [113,114,115]. Meanwhile, the continuous inflammatory responses triggered by chronic endogenous hypoxic in WAT inhibit the differentiation of preadipocytes into mature adipocytes, which impairs the capacity of WAT lipid storage [116]. This vicious cycle will greatly elevate the level and duration of hypoxic. Therefore, hypoxic and inflammation reinforce each other in promoting adipose tissue dysfunction [117].

In addition, maintenance of the immune system accounts for 20% of the energy expenditure in total body [118]. To preferentially ensure sufficient energy utilization in immune cells, immune cells secrete inflammatory cytokines, such as IL-1β, monocyte chemoattractant protein-1(MCP-1), macrophage galactose-type lectin 2(Mgl2) and TNF-α that downregulates insulin signaling to reduce glucose utilization in non-immune systems [119,120]. This suggests that hyperactivation of immune systems may contribute to the development of systemic insulin resistance in obesity. Moreover, hypoxic-induced inflammation downregulates the expression of PPARγ [112]. In fact, inflammation-induced TNF-α downregulates PPARγ gene expression and transcriptional activity in adipocyte [121]. A decrease in PPARγ abundance will upregulate TGF-β1 levels [122]. Notably, white adipocytes of obese mice release abundant TGF-β1, which directly upregulates the expression of myofibroblast gene expression (e.g., Acta2 and Tagln) in myofibroblasts of WAT through the activation of downstream Smad3, and it eventually induces WAT fibrosis. [123,124].

At the same time, some studies have also revealed that the responses of adipocytes to various oxygen concentrations were different when the cultured human primary adipocytes were exposed to physiological hypoxic. The decreased responsiveness of adipocytes to inflammatory stimuli (e.g., TNFα) under hypoxic may represent an intrinsic adaptive mechanism to maintain adipose tissue function under hypoxic and inflammation [125]. Meanwhile, lower pO2 reduced pro-inflammatory gene expression and improved the metabolic phenotype of differentiated human adipocytes, and had a greater effect on adipokine secretion [126]. While O_2_ levels in the human body range from 3% to 11%, change in oxygen levels during adipogenesis significantly contributes to adipocyte function, such as regulating the release of adipokine and especially adiponectin, as well as hormone-induced lipolysis pathway.

The intervention of 1% oxygen concentration on adipocytes cannot perfectly recapitulate the physiological condition under which adipocytes are exposed to hypoxic. Collectively, these events induce acute hypoxic in adipocytes, dysregulation of adipokine secretion, increase in inflammatory factors and decrease in adiponectin release [127]. The above studies, in terms of the hypoxic exposure, can better reflect the physiological relevance in the human when compared to the acute exposure of (pre)adipocytes to severe hypoxic in an acute setting, which are conducted in vitro.

##### Fibrosis Contributes to the Dysfunction of White Adipose Tissue Induced by Hypoxic

WAT fibrosis is a pathological phenomenon caused by excessive deposition of ECM proteins, and it is a hallmark of metabolically dysfunctional adipose tissue [128]. Myofibroblasts play an important role in the progression of fibrosis as they produce excessive ECM proteins in WAT [129,130]. The ECM is composed of various components, including collagens, fibrillins, proteoglycans and non-proteoglycan polysaccharides [131,132]. During the progression of obesity, adipocyte progenitors can differentiate into myofibroblasts and synthesize collagen [130,133]. HIF-1α activation by hypoxic upregulates the expression of ECM proteins, such as Lysyl oxidase (LOX), collagen I and III in WAT, which would further drive interstitial fibrosis in adipose tissue [128,134]. Increased ECM stiffness prevents adipocytes from adaptive expansion and leads to adipocyte death, decreased lipolysis and fatty acid extravasation [135,136]. The chemotaxis induced by HIF-1α recruits proinflammatory cells, including M1 macrophages and monocytes to WAT [134,137]. This in turn exacerbates the severity of hypoxic and inflammatory stress in WAT. Meanwhile, WAT fibrosis decreases lipid storage capacity in adipocyte and increased FFA levels [138,139]. Excess FFAs also activates Toll—Like receptor 4 (TLR4) in macrophages and recruits the macrophage-inducible C-type lectin (CLEC4E), which is essential for the expression of fibrosis-related genes and regulates obesity-induced adipose tissue fibrosis [140].

Furthermore, inflammatory responses and fibrosis also diminish angiogenesis of adipose tissue. VEGF-A_165_b is an inhibitory isoform of VEGF-A, and inflammation-driven VEGF-A_165_b expression induces dysfunction in adipose tissue angiogenesis in obese humans [135,136]. Hypoxic may activate downstream Wnt signaling via HIF-1α, and WNT5A is known to regulate adipose tissue angiogenesis via antiangiogenic VEGF-A_165_b [136,141]. Previous studies also reported the potential regulation of angiogenesis by ECM stiffness [142]. Obesity contributes to the inhibition of TWIST1-SLIT2 signaling and impaired angiogenesis, while Twist1 acts as a mechanosensor that senses ECM stiffness [143,144]. Recent research reported the positive effect of HIF-1α/Notch-1/PDGFR-β/YAP-1/Twist-1 axis in promoting the fibrotic phenotypes of pericytes [145]. Consequently, adipose tissue fibrosis may suppress angiogenesis through blockade of the HIF-1α-TWIST1 signaling axis. However, the molecular mechanisms in inhibition of angiogenesis in obese adipose tissue and the roles of endogenous hypoxic are still elusive, and further investigation are warranted.

##### Hypoxic Causes Adipose Tissue Dysfunction Accompanied with Ectopic Lipid Accumulation

The dysfunction in adipose tissue reduce the capacity of nutrient storage in adipose tissues. And abundant FFAs are released into the systemic circulation, causing ectopic lipid accumulation in tissues and organs like liver, skeletal muscle, and heart, and eventually elicits lipotoxicity inside these organs [138]. Lipotoxicity widely impacts the function of various organs and it systemically disturbs metabolic homeostasis. For example, NAFLD is frequently associated with obesity, as abnormal hepatic lipid metabolism give rise to increased synthesis of lipotoxic intermediates (e.g., diacylglycerol) that induced insulin resistance, local tissue inflammation, hepatocellular damage, NASH, and fibrosis in the liver [146,147].

##### Hypoxic Induces Mitochondrial Physiological Adaptation and Leads to Lipid Accumulation

Mitochondria play the central role in governing cellular energy metabolism and transduction of stress signals [148,149]. However, the role of mitochondrial function in WAT is still less delineated than that in brown adipocytes. Recently, the role of mitochondria in the maintenance of WAT function and systemic metabolic health has been appreciated [91,150]. Glucose uptake and utilization by adipocytes are highly sensitive to exogenous O2 concentration [151]. And mitochondria act as cellular O2 sensors mainly through sensing a series of hypoxic stress signals elicited by reactive oxygen species (ROS) and HIF-1α [152]. Hypoxic directly affects glucose metabolism pathways in white adipocyte mitochondria, as deficiency in O2 concentration impairs the ability of aerobic glycolysis in mitochondria [153,154]. Therefore, glucose metabolism in adipocyte gradually switches from aerobic glycolysis to anaerobic glycolysis in response to hypoxic [153].

HIF-1α activates the mitochondrial glycolytic pathway by up-regulating the expression of phosphoglycerate kinase 1 (PGK1) and lactate dehydrogenase A (LDHA), as well as priming adipocytes to selectively increase expression of glucose transporters (e.g. GLUT 1) and to enhance monosaccharide (e.g., glucose) uptake to compensate the impairment of ATP production by anaerobic glycolysis compared to aerobic metabolism does [155]. Thereby, the lack of oxidative phosphorylation capacity caused by hypoxic is ameliorated [155,156]. Meanwhile, lactate production is increased in parallel with the HIF-1α-dependent upregulated expression of lactate transporters MCT1 and MCT4, which belong to the family of proton-linked monocarboxylate transporters (MCTs) [153]. Enhancement of lactate production has important pathological implications in obesity. As a fulcrum of metabolic regulation, lactate inhibits lipolysis via binding to and activating the Gi-coupled receptor 81 (GPR81) in adipocytes, and this process may regulate the downstream cAMP-PKA pathway to influence mitochondrial β-oxidation [157,158]. Meanwhile, elevated lactate upregulates the intermediate malonyl-CoA, which reduces fatty acid uptake by muscle mitochondria via inhibiting carnitine-palmitoyl transferase-1 (CPT1) [159]. These effects in both directions of lipid metabolism collectively lead to intracellular lipid accumulation. Furthermore, as glycerol synthesis requires intact mitochondrial function, dysfunctional mitochondria may affect the synthesis of triacylglycerol in adipose tissue, decreasing the storage of triacylglycerol in adipocyte lipid droplets [160].

In conclusion, endogenous hypoxic effectively triggers inflammation, ECM remodeling and mitochondrial dysfunction. And hypoxic-induced systemic inflammation further exacerbates insufficient vascularization and tissue fibrosis in WAT. This transition process will aggravate adipose tissue dysfunction. Moreover, the integration of augmented secretion of inflammatory cytokines, altered mitochondrial metabolism and ectopic lipid deposition negatively blunt insulin signaling, coordinate systemic insulin resistance and increase probability of metabolic diseases and their complications [161,162,163,164]. Interestingly, recent studies have highlighted the molecular and functional heterogeneity of depot-dependent adipocyte progenitors [69]. Upon hypoxic stimulation, whether different subpopulations of adipose progenitor cells have unique changes in adaptation, the subsequent distinct signaling pathways, and their functions in promoting obesity need to be further investigated. Additionally, the time-oxygen concentration relationship between hypoxic and white adipose tissue dysfunction remains poorly understood.

### 3.2. Endogenous Hypoxic and Brown Adipose Tissue

There were few reports to elucidate the effect of obesity-induced local tissue hypoxic on the structural and functional alterations in BAT. Notably, hypoxic potentially triggers BAT “whitening” [165,166]. BAT in obese animals has diminished rate of lipolysis, which is in contrast to obesity-driven adaptive hypertrophy of adipocyte. This adverse trend reduces the O2 tension between the BAT and its surrounding tissues in the obese population. However, BAT is a highly vascularized tissue with abundant mitochondria and sensitively perceives O2 concentration.

Nevertheless, the high intracellular fatty acid level and hypoxic microenvironment caused by obesity synergistically inhibit β1 and β3 adrenergic signaling. β-adrenergic signaling is the major pathway to promote *Vegfa* expression in brown adipocytes [167]. Therefore, decreased expression of *Vegfa* in obese animals reduces blood vessel number and density and causes insufficient oxygen supply in BAT, which further contributes to the accumulation of large lipid droplets and mitochondrial dysfunction, thereby inducing the BAT “whitening” [166].

Importantly, the development of obesity and its complications is largely dependent on the abnormal functioning of BAT [52,168]. BAT regulates organismal energy homeostasis via non-shivering thermogenesis and secretion of batokines [169,170]. A series of adverse events such as reduced BAT thermogenesis, brown adipocyte death and tissue inflammation occur during the whitening process [165]. Combined with WAT dysfunction, the systemic metabolic disorder and abnormally organized adipose tissue caused by obesity are further aggravated.

Increasing the activity of BAT in vivo can protect against metabolic syndrome [171]. Mitochondria in BAT play a key role in BAT to systemically regulate metabolic homeostasis. Mitochondria of BAT play a key role to systemically regulate metabolic homeostasis. Notably, the defective catabolism of branched-chain amino acid (BCAA) in mitochondria of BAT leads to the development of diet-induced obesity and glucose intolerance. Conversely, systemic glucose homeostasis is improved by enhanced BCAA catabolism in mitochondrial. [172]. However, obesity-induced endogenous hypoxic significantly increases mitochondrial ROS (mtROS) production [166], which is associated with mitochondrial membrane depolarization and oxidative stress [173]. This subsequently alters the expression of proteins related to mitochondrial dynamics, such as downregulation of mammalian mitochondrial fusion related proteins optic atrophy 1 (OPA1), models 1 and 2 (Mfn1 and Mfn2), thereby impairing mitochondrial function in brown adipocytes [174,175,176]. The dysregulation of mitochondrial dynamics in brown adipocytes in obesity is worthy of more investigation.

Furthermore, hypoxic also significant increases the expression of both PTEN-induced putative kinase protein 1 (PINK1) and Parkin, and promotes the activation of a mitophagic program in BAT, which suggests that the process of mitophagy is induced [166]. The reduced content of mitochondria leads to decreased substrate oxidation, and it aggravates lipid accumulation in BAT [165,177]. On the other hand, the BAT-specific deletion of Atg7(an essential gene for autophagy) confers the mice resistance to diet-induced obesity, featured with elevated mitochondrial content, improved glucose metabolism and reduction in body weight [178]. Collectively, these results suggest that obesity regulates hypoxic stress in mitochondria, thereby promoting the BAT dysfunction. Moreover, based on the inhibitory effect of endogenous tissue hypoxic on mitochondrial biogenesis and function in BAT, hypoxic is one of the putative key factors that promote the transition of BAT to white adipose tissue and further mechanistic investigation is warranted.

## 4. Targeting the Critical Mediators in Endogenous Hypoxic to Combat Obesity and Comorbidities

As we summarized in Section 3, multiple factors mediate locally endogenous hypoxic in adipose tissues and contribute to their dysfunction. Given the fact that local hypoxic in adipose tissue, as well as lower oxygen consumption (albeit under certain circumstances it has higher oxygen pressure in people with obesity) during the progression of obesity may contribute to obesity, normoxia in adipose tissues may attenuate the inflammation, fibrosis and dysregulated metabolic traits. Here we will discuss some of the potential therapeutic targets and small molecules that may ameliorate endogenous hypoxic by inhibiting the functions of these effectors.

### 4.1. Lysyl Oxidase (LOX) Inhibitors

As we discussed in above section, HIF-1α activation by hypoxic upregulates the expression of lysyl oxidase (LOX), collagen I and III in WAT. The lysyl oxidase (LOX) family of enzymes is consisted of five members and is pivotal in the maintenance of collagen deposition and stability. They oxidize the lysine residues in the side chain of collagen and elastin to generate aldehyde and the aldehyde crosslinked with other collagens [179,180]. Excessive crosslink of collagens eventually increases matrix stability and develops into fibrosis. The detrimental effect of LOX in mediating fibrosis of adipose tissues suggests it as a potential target that may be blocked to partially reverse the effect of endogenous hypoxic in adipose tissues.

Chaudhari et al have reported a class of LOX inhibitors that covalently modified and irreversibly inhibited lysyl oxidases. The inhibitor, PXS-6302 effectively blocked the excessive crosslinking of collagens and ameliorated the skin scarring and fibrosis in murine models [181]. Moreover, pan- Lysyl oxidase inhibitor PXS-5505 attenuated multiple-organ fibrosis in murine models [182]. Thus, it is feasible to validate their anti-fibrotic activities in adipose tissues upon endogenous hypoxic, as a novel therapeutic alternative to combat obesity.

### 4.2. Antagonists of LTB4 Receptor

The state of endogenous hypoxic in obesity induces HIF-1α and the secretion of LTB4 [80]. Arachidonic acid undergoes the enzymatic oxidation process to generate Leukotriene B4 (LTB4) and LTB4 is released from the cell membrane [183]. It binds to the leukotriene B4 receptors, BLT1 and BLT2 and recent advances have unraveled the function and structures of BLT1 and BLT2 in mediating inflammatory responses [184,185], as well as BLT1 in hepatocytes promotes NAFLD development in obesity [186]. Therefore, design of antagonists against BLT1 and 2 based on the crystal structures of BLT-ligands to block the downstream inflammatory pathways may be a promising strategy to treat obesity and its comorbidities. 

Indeed, BLT antagonists have exhibited anti-inflammatory activities in multiple experimental inflammation models, such as murine pemphigoid disease [187], experimental periodontitis [188]. It is feasible to conduct structure-activity relationship studies to identify more potent antagonists to diminish inflammation caused by endogenous hypoxic. Probing the potential activities of BLT and LOX inhibitors *in vivo* warrants future investigation to block endogenous hypoxic.

## 5. Exogenous Hypoxic and Obesity

In contrast to endogenous adipose tissue hypoxic caused by obesity, exogenous hypoxic has some positive effects in combating obesity and the underlying mechanism is still under investigated. Moreover, different levels of low oxygen function differentiate in the capability to resolve obesity and will be intensively discussed in this section.

### 5.1. The Positive Effect of Exogenous Hypoxic

Dünnwald et al. [189] conducted quantitative meta-analysis of changes in body weight and body composition that were influenced by different altitudes. The altitudes were classified using widely accepted criteria (moderate altitude: 1500–3500 m, high altitude: 3500–5300 m, extreme altitude: >5300 m). The results showed that physical activity under hypoxic conditions had a positive effect on managing normal body weight and improving body composition, and the effect was further amplified with the increase in exposure time to hypoxic. Among them, moderate altitude exposure may have higher efficacy in the treatment of obesity. High altitude may reduce weight by reducing appetite, increasing energy expenditure and causing a negative energy balance [189,190]. Voss et al. [191] also showed that the prevalence of obesity in the USA was inversely associated with altitude and urbanization after adjustment of temperature, diet, physical activity, smoking history and demographic factors. Meanwhile, several studies have shown that lifetime or long-term (months to years) exposure to high altitude is associated with lower risk of obesity and weight loss [192,193,194,195]. These findings revealed the putative and potential beneficial effects of exogenous hypoxic intervention on obesity and body composition.

Exposure to exogenous hypoxic has a positive effect on people with obesity and enhances energy expenditure in sedentary, overweight men [196,197]. Workman et al. [197] demonstrated that six of 3-h passive exposures to hypoxic (blood oxygen saturation (PO_2_) = 80%) increased energy expenditure and lipid oxidation in overweight men compared with those exposed to normoxia. In six overweight or obese individuals, Costalat et al. [198] showed that 10 episodes of 70-min intermittent hypoxic (approximately 15 episodes of setting oxygen to PO_2_ = 80%) over 2 weeks reduced lipid levels (Lipoprotein cholesterol, LDL-C) and lowered systolic blood pressure. Marlatt et al. [199] evaluated significant improvements in glucose tolerance after 14 consecutive nights of hypoxic exposure (15% O_2_), which only occur at night (sleeping in a hypoxic tent at home) in patients with obesity and type II diabetes. These pilot studies showed that intermittent hypoxic exposure (i.e., from 70 min to 12 h every 24 h) improved glucose tolerance in individuals with type II diabetes for a relatively short period of time (1-2 weeks) and without additional intervention (nutrition or physical activity). While acute exposure to hypoxic disrupts metabolic homeostasis, long-term exposure (2 weeks) may normalize glucose homeostasis and even improve glucose disposal in the case of obesity [200]. In several cohort studies of exposure with 15% oxygen concentrations, exposure to hypoxic reduces the risk of obesity [201], increases the rate of lipid oxidation [202,203] and decreases serum triglycerides levels [202].

### 5.2. Hypoxic Caused Negative Energy Balance

The most effective strategy for weight loss is to increase energy expenditure, while suppressing excessive appetite [204]. Reduction in food intake without exposure to hypoxic usually results in a substantial decrease in resting metabolic rate [205,206,207]. Oltmanns et al. [208] found that when 13 non-obese men were exposed to hypoxic for 30 min by decreasing oxygen saturation to 75% (vs 96% in a control session), those adults acutely exposed to hypoxic had elevated their resting metabolism rate. The enhancement may be associated with increased cardiopulmonary function, enhanced sympathetic nerve activity, upregulated circulating IL-6 levels and increased thyroid activity. Westerterp et al. [209,210] observed the negative energy balance in the human body at altitudes of 4350–5000 m, probably due to increased energy expenditure as initial hypoxic-induced loss in body mass was higher than that of later on during the experiment. This may further corroborate the conclusion that hypoxic exposure is the paradigm for successful weight loss.

### 5.3. Adverse Events of Obesity Due to Hypoxic

Nevertheless, exposure to hypoxic may have some adverse consequences, which depend on the oxygen concentration and duration of hypoxic exposure. For instance, in 9 of non-obese men, each experimental trial was conducted with a basal metabolic rate (BMR) measurement as baseline. It was followed by constant workload exercise performed under normoxia or in moderate hypoxic (12% oxygen concentration), followed by a resting metabolic rate measurement 40–60 min post-exercise (PEMR40–60). It revealed that plasma FFAs and triglycerides increased after 60 min intervention [211]. Moreover, induced chronic intermittent hypoxic (CIH) (sleep apnea, a hallmark of obstructive sleep apnea, OSA) [212,213] has aggravated insulin resistance in obese mice and promoted steatohepatitis, thereby exacerbating obesity [214,215,216,217,218]. As most of these studies have used hypoxic to mimic the status of pathological hypoxic, chronic intermittent hypoxic caused by OSA in obese patients may be one of the underlying mechanisms of the obesity-morbidities paradox [219]. These findings suggest the concentration and duration of hypoxic in patients with sleep apnea syndrome should be critically assessed for the appropriate interventions.

### 5.4. Mechanism of Different Hypoxic Concentration on Energy Metabolism

#### 5.4.1. Exogenous Oxygen Concentrations Exceed 10%

Females exposed to 15% hypoxic had approximately 20% less gonadal white adipose tissue during mid to late gestation compared to females exposed to normoxia. Hypoxic (15% hypoxic) mothers improved glucose tolerance and insulin sensitivity compared to normoxic mothers and did not develop insulin resistance in late pregnancy, which was associated with upregulation of HIF-1α and its target genes, which in turn increased glycolysis [220]. After 8 weeks of intervention with 14.07% hypoxic exposure, plasma cholesterol and triglyceride content were significantly reduced, along with improved insulin sensitivity, and the area and diameter of visceral adipocytes in the hypoxic group were significantly smaller than those in normoxia group. Meanwhile, genes in mitochondrial biosynthesis, such as peroxisome PGC-1α, nuclear respiration factor 1(NRF1) and mitochondrial transcription factor A, were upregulated [221,222,223]. Ge et al. [224] found that high-altitude exposure induced a significant increase in BAT activity in subjects at an altitude above 3000 m (approximately 14.4% oxygen concentration). Adipose-derived exosomal miR-210 modulated insulin resistance and glucose metabolism, along with BAT activation in humans and rodents [225]. For example, miR-92a is associated with high-density lipoprotein (HDL) components and indicates an increased risk of cerebrovascular disease (CVD) [226]. Preclinical studies demonstrated that attenuation of exosome mir-210 secretion could enhance the expression of fibroblast growth factor receptor 1 (FGFR-1) in BAT, and promote energy expenditure [224,227]. Transcriptomic data from 11.2% oxygen exposure showed that 802 genes were significantly upregulated and 1755 genes were downregulated in the BAT of obese mice compared to normal mice. The metabolic pathways of these genes were mainly enriched in glucose and lipid metabolic processes and inflammatory responses. Hypoxic exposure regulates BAT mainly through HIF-1α, phosphatidylinositol 3-kinase/protein kinase B pathway (PI3K-Akt), forkhead Box O (FoxO) and epidermal growth factor receptor (ErbB) signaling pathways [228]. Ge et al. [229] demonstrated that high-altitude induced hypoxic may rescue mitochondrial function and activate adenosine 5′-monophosphate (AMP)-activated protein kinase(AMPK) signaling in obesity-induced fatty liver in mice. Song et al. [229] found that mice at a simulated 4500 m altitude (oxygen concentration of approximately 11.8%) could improve body composition by activating mitochondria biogenesis, and reduce enzymes protein expression involved in lipid synthesis and increasing lipolysis.

#### 5.4.2. The Oxygen Concentration of 5% to 10%

Mice in a high-fat diet (HFD) -induced overweight/obesity model were exposed to 10% oxygen for one hour per day and hypoxic counteracted weight gain and insulin resistance induced by HFD. The HFD-induced expansion of white and brown adipocytes and fatty liver were reversed and normal liver function was restored. Hypoxic induces expression of factors such as UCP1, ADR3 (β3-adrenergic receptor), CPT1A, adipose triglyceride lipase (ATGL), PPARα and PGC1α, and arginase in liver. Moreover, the M2 macrophage marker, CD206 and recombinant PPARγ in liver and and UCP1 in brown fat were upregulated [230]. In contrast, in ApoE knockout mice, 6.5% hypoxic intervention increased adipose Angptl4 levels, inhibited adiponectin lipase, increased fasting plasma triglyceride and very low-density lipoprotein cholesterol levels and increased the size of atherosclerotic plaques [231].

These results showed that different oxygen concentrations induced divergent molecular adaptive changes, as shown in Figure 2, which explained the mechanism of adipose tissue on metabolism under different oxygen concentrations and provided new evidence for identifying optimal oxygen concentrations for the treatment and prevention of obesity.

## 6. Application of Hypoxic Exercise in Treatment of Metabolic Disease

### 6.1. Obesity

A combination of exercise and hypoxic exposure is proposed to be a promising complementary treatment for obesity [232,233,234] as exercise effectively ameliorates obesity and improves systemic metabolism [235,236,237]. Mackenzie et al. [238] showed that when patients with type II diabetes, overweight, or obesity underwent a 60-min exercise test under hypoxic conditions (an inhaled oxygen fraction of 0.146), the glucose tolerance and peripheral insulin sensitivity were improved within the following 4 h as compared with patients who underwent normoxia exercise. In people with obesity, hypoxic training reduced triglyceride levels more significantly than subjects with normoxia training [31,239]. This suggested that repeated exercise under hypoxic conditions (e.g., training) may provide additional benefits for obese individuals compared with normoxia exercise training did. However, optimal interventions for obese people at different concentrations of oxygen may vary among different subjects, as shown in Table 1. The oxygen concentration between 14–15% was an optimal range of oxygen concentration for curbing obesity and promoting weight loss. The frequency of intervention was 4–8 weeks, 3–6 times a week, and each intervention took 60 min. Mechanistic studies indicated that hypoxic training increased activities of glycolysis-related enzymes, mitochondrial number and GLUT-4 levels, and improved insulin sensitivity [240]. Four weeks of intermittent hypoxic training at 11.2% oxygen concentration significantly increased the expression of nuclear receptor PPARα and activated AMPK-related pathways in an AMPKα2 isoform-dependent manner [241,242]. At 14.7% oxygen concentration, hypoxic training significantly inhibited the hyperactivation of the hepatic endocannabinoid system (ECS) and mitigated hepatic steatosis, presumably by down-regulating the CB1/SREBP-1/PPARγ signaling pathway in obese mice [243,244]. These findings also provided a new paradigm for hypoxic training in the treatment of obesity. Most of the intervention schemes of hypoxic training on obese people were performed within 4–8 weeks. It may be due to the reason that adaptation of the hypoxic environment of the body is different from the acute stimulation caused by the short time (several hours).

### 6.2. Diabetes

Obesity is intimately associated with hyperinsulinemia and insulin resistance, which are the hallmarks of diabetes [245]. Recent studies suggest that hypoxic exposure may normalize global glucose homeostasis, in a randomized, controlled, single-blind crossover study of the exposure of 12 overweight/obese men to 15% O_2_ for 7 days. It unraveled no significant changes in plasma glucose and insulin concentrations, nor in insulin sensitivity of adipose tissue, muscle and liver. Nevertheless, it indicated that hypoxic exposure increased insulin-independent glucose uptake in human primary myotubes through activation of AMPK. [246]. Meanwhile, Lecoultre [247] et al. found that exposure to moderate hypoxic (15 ± 0.5% O_2_, ~2400 m altitude) for 10 consecutive nights in obese men (a within-group comparison without controls) increased insulin sensitivity. Some studies also found that for type 2 diabetes patients, peripheral insulin sensitivity increased during hyperbaric oxygen treatment (HBOT), and this increase was maintained for 30 min after HBOT [248,249,250].

The possible contradiction with hypoxic intervention is that this study included 8 diabetic patients in the community with the limited sample size. Meanwhile the other studies included either an acute intervention for 2 h, or only 5 diabetic patients were recruited. Therefore, more clinical trials are needed to verify the conclusions of hypoxic in the treatment of diabetes.

Moreover, most of the studies with hypoxic exposure or hypoxic training were conducted under normobaric pressure. Whether hyperbaric oxygen interventions cause distinct alteration in body adaptability remains to be further discussed. Veerle van Hulten et al. reviewed the effects of hypoxic exposure on human glucose homeostasis, which is metabolically impaired. Hypoxic exposure improves glucose homeostasis, whereas hypoxic exercise training (2–8 weeks) appears to have no additional effect on glucose homeostasis compared with normoxic exposure. Due to the heterogeneity in the tested people and duration of intervention (acute study/2–8 weeks of training), it is difficult to identify the critical factors that explain the conflicting findings. At the same time, the deficiency in control groups in some studies also affected the interpretation of their results [251].

In summary, hypoxic exposure under resting and exercise conditions may provide a novel therapeutic strategy to improve glucose homeostasis in metabolically impaired individuals. Yet more large-scale randomized controlled trials are necessary to draw reliable conclusions on the effects of hypoxic exposure for glucose homeostasis.

As a combination of different forms of exercise, hypoxic training also plays a certain role in treatment of diabetes. The study showed an increased oral glucose tolerance after acute 14% hypoxic intervention, which may be closely related to exercise load or age mismatch between the two groups of subjects (healthy and prediabetic) [252]. A study by Żebrowska et al. found that cycling at 15.2% oxygen concentration decreased levels of blood glucose and might produce significant benefits in preventing cardiovascular complications in diabetes. Meanwhile, some studies have found that acute hypoxic combined with exercise can improve insulin sensitivity, which may be related to the intensity and duration of exercise [238,253].

Potential adaptations to hypoxic effects may vary with the severity and duration of hypoxic exposure (duration of each hypoxic event and total exposure duration) and the exposure pattern (intermittent/continuous) [254]. It has been proposed that mild hypoxic exposure with low cycle numbers (3–15 cycles/day) may improve several metabolic parameters, whereas frequent episodes of severe hypoxic may lead to pathological adaptation [255]. For example, severe (intermittent) hypoxic exposure may increase the risk of sympathetic nervous system activity, blood pressure, inflammation, cholesterol levels, atherosclerosis and right ventricular hypertrophy, and impair cognitive function [255].

## 7. Discussion and Future Direction

### 7.1. The Distinct Features of Endogenous and Exogenous Hypoxic

The endogenous hypoxic associated with adipose tissue expansion, inflammation and cell proliferation during obesity is probably an unappreciated important mediator in promoting adipose tissue dysfunction. Adipose tissue exhibits adaptive adjustment in intrinsic structure and cellular function when exposed to endogenous hypoxic for a short period of time. On the other hand, continuous adipose tissue expansion due to chronically excessive energy will lead to the transition from “acute” hypoxic to “chronic” hypoxic. The dysfunctional adipose tissue releases a myriad of pro-inflammatory factors and lipotoxic molecules, which in turn exaggerates obesity.

The following questions remain to be addressed:How to define “acute” and “long-term” hypoxic periods;Given the complexity in the physiological conditions of organisms, previous experiments may not adequately mimic the endogenous hypoxic in adipose tissue.

Therefore, novel approaches that recapitulate the physiologically relevant hypoxic environment are critical for future interrogation. Notably, 1% O_2_ was commonly used in studies of cellular response to hypoxic and creates an extreme hypoxic state for adipocytes, which can only recapitulate an acute stimulus or an emergent response of adipocytes under severe hypoxic [9]. The partial pressure of oxygen in the white adipose tissue of obese mice was as low as 15 mmHg, which was equivalent to 2% O_2_, as measured in previous studies [9]. On the other hand, PO_2_ in adipose tissue of lean mice were between 45 to 50 mmHg, which was similar to the general level of oxygenation in tissues (~6.5% O_2_) [78,256]. O_2_ directly modulates gene expression and secretion of leptin and VEGF in a dose-dependent way [257]. Their mRNA abundance and secretion increased significantly when PO_2_ was reduced. For other adipokines, e.g., leptin, the peak O_2_ concentration for stimulation of their mRNA and secretion was at 5% [9]. Adipocytes were highly sensitive to O_2_ concentrations, while their metabolic functions did not fluctuate significantly under the physiological range of O_2_ concentrations. Thus, the caveats are that external hypoxic concentrations did not reflect the actual oxygen concentration of adipose tissue. Therefore, more considerations are needed in terms of the oxygen sensitivity of the actual adipose tissue, including the overall effects of other variables, such as external temperature, stress and UV exposure.

Notably, the discrepancy between endogenous hypoxic and hyperoxia in adipose tissues revealed by multiple landmark studies may be attributed to the complex genetic or adaptive etiology of obesity, which leads to different oxygen concentrations in adipose tissues. Thus, the effect of endogenous hypoxic or hyperoxia in adipose tissues should be critically assessed in large-scale trials that maximize the coverage of obese population with different genetic traits.

In addition, the discrepancy between the metabolic effects induced by endogenous and exogenous hypoxic may be explained by the fact that hypoxic exerts the metabolic effect on other tissues and organs throughout the body, thereby indirectly affecting adipose tissue metabolism. For example, hypoxic exposure increases whole-body energy metabolism [258], while longer exposure to high altitudes appears to increase the catabolic capacity of glucose in skeletal muscle [240,259], and intermittent hypoxic exposure improves cardiovascular function and increases glucose utilization in patients with Type 2 Diabetes [260]. The hypoxic effect of multiple tissues and organs may be responsible for the overall weight loss in people with obesity, which overrides the obesity-promoting effects of endogenous hypoxic on adipose tissues. Hypoxic exercise partially alleviates the adverse effects caused by exposure to pure hypoxic, and combination of exercise and hypoxic conditions synergistically normalize glucose and lipid metabolism, increase blood flow and resolve inflammation and fibrosis.

### 7.2. The Translational Perspective of Combining Therapies to Target Endogenous and Exogenous Hypoxic

We summarized the effect of hypoxic on adipose tissue, and discussed the latest progress and intervention plans to treat obesity. Notably, the article characterized the changes of various factors under different oxygen concentrations. We proposed that the difference between exogenous and endogenous hypoxic was responsible for the different molecular alterations in adipose tissue.

In terms of translational perspective, currently no targeted therapies have been reported to effectively and durably reverse the pathological expansion of adipose tissue, inflammation, fibrosis, etc., in order to improve adipose tissue health in obesity and metabolic diseases. Nevertheless, endogenous and exogenous hypoxic can be integrated and exploited to generate synergistic effects in treatment of obesity and comorbidities. Integration of endogenous and exogenous hypoxic under the context of obesity progression enables a deeper understanding of the mechanism and development of interventions. For instance, the molecules that regulate the activity of the mediators in endogenous hypoxic in Section 4 can be combined with hypoxic exercise to augment the beneficial effects of exogenous hypoxic and repress the endogenous hypoxic simultaneously. The combination of the therapies will offer promising alternatives to treat obesity and comorbidities.

## Figures and Tables

**Figure 1 cells-11-03735-f001:**
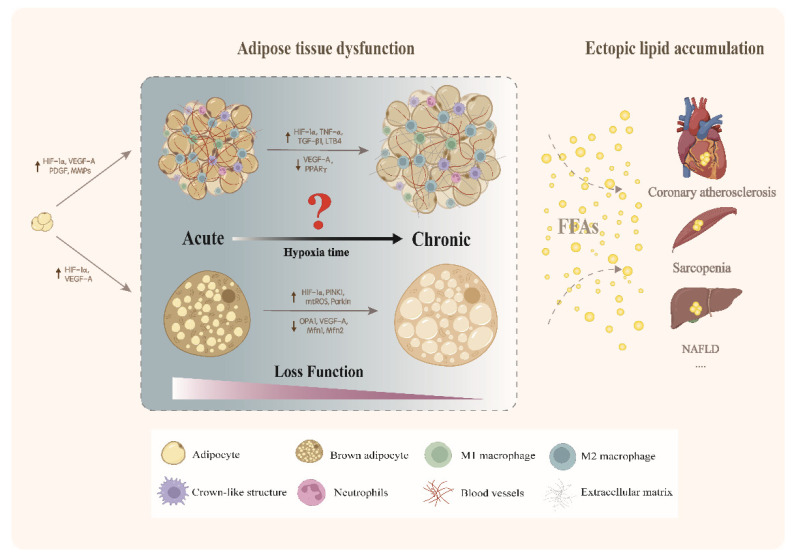
Adipose tissue adaptively remodels to cope with obesity (e.g., cell proliferation and expansion, increased energy metabolism). Angiogenesis is increased to counteract the acute endogenous hypoxic that accompanies adipocyte expansion. With the development of obesity, chronic endogenous hypoxic will induce inflammation, fibrosis of WAT and BAT “whitening”, leading to ectopic lipid accumulation in other organs. This may eventually give rise to metabolic diseases, such as coronary atherosclerosis, sarcopenia and NAFLD.

**Figure 2 cells-11-03735-f002:**
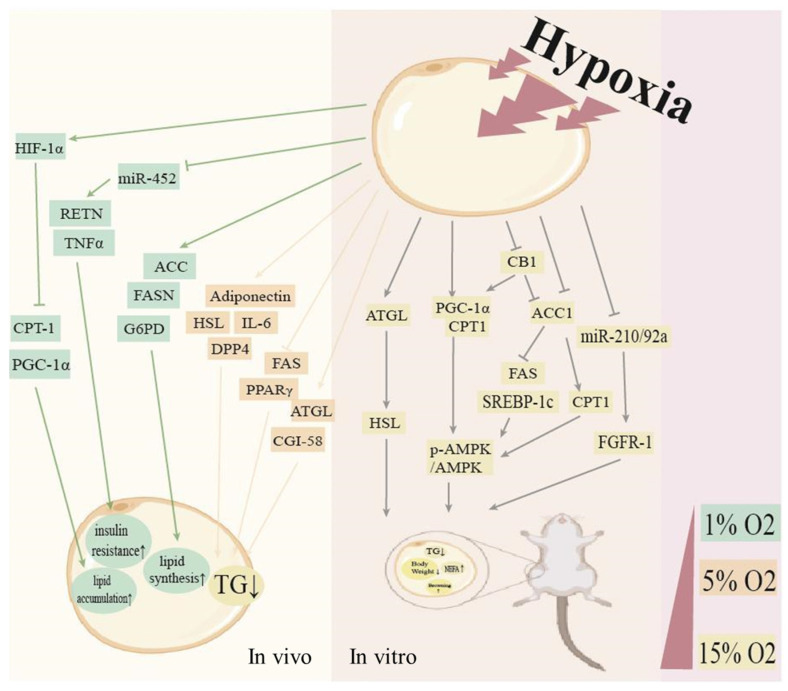
Changes in adipocyte pathways under different hypoxic interventions. In in vitro studies of cellular responses to hypoxic, 1% O_2_ is usually used. This was considered as severe hypoxic, as well as acute irritation to the body. However, this was only an extreme case of the mechanism being used to prove. Usually, the body will adapt to the severe hypoxic, and, meanwhile, different concentrations of O_2_ in adipose tissue will induce changes in different signaling pathway with distinct outcomes. Different colors represent different oxygen concentrations.

**Table 1 cells-11-03735-t001:** Intervention effects in obese patients under different hypoxic conditions.

Subject	Country	Age	FiO_2_	Hypoxic Mode	Exercise Method	Result	Reference
23 obese Individuals	France	52 ± 12 year	13%	LoHi	8 weeks, 3 day/week, 75% VO_2_ max	DBP↓, body composition no change	[184]
24 Korean obese men	Korean	66.5 ± 0.8 year	14.40%		12 weeks, 3 day/week, 30 min, aerobic exercise + resistance exercise 30–40 min	Body weight↓, BMI↓, Fat mass↑, Lean mass↑	[186]
32 Korean obesity women	Korean	47.5 ± 7.5 year	14.50%		12 weeks, 3 day/week, 50 min/d, Pilates	DBP↓, TC↓, TG↓	[187]
20 sedentary subjects	Japan	30 ± 2 year	15%		4 weeks, 3 day/week,1 h/d, 55% VO_2_ max	Glucose tolerance↑, Body Weight and abdominal fat area no change,	[189]
14 obese adolescents	Belgium	12–15 year	15%		6 weeks, 3 day/week, 50–60 min/d, endurance and resistance exercises	TG↓, glucose levels↓, AUC of insulin↓	[179]
Overweight to obese	Germany	42.2 ± 1.2 year	15%		4 weeks, 3 day/weeks, 65% VO_2_ max	Fat-free mass↑, Waist circumference, Fasting insulin↓, Fat mass↓	[193]
59 overweight/obese women	Uruguay	-	17.20%		12 weeks, 3 day/week, High intensity exercise	Waist circumference↓, percentage of trunk fat mass↓,	[195]
49 obesity male Individuals	Spain	20–50 year	13.7–16.7%	IHT	8 weeks, 2 day/week, 1 h/d,50% aerobic,50% strength	Body weight↓, BMI↓, waist circumference↓	[185]
14 obese adolescents	Belgium	12–15 year	15%		30 weeks, 1 day/weeks, 50–60 min/d, 12 min bicycle and strength training of the abdominal, quadriceps and biceps muscles	Body weight↓, Fat mass↓,	[190]
23 obese men	Germany	52.3–62.5 year	15%		6 weeks, 3 day/week, 1 h/d, 60% VO_2_ max	BMI↓, Fat mass↓, Lean mass↓, HDL↑, TG↓	[191]
32 obese people	Germany	mean age 45.5 year	15%		8 weeks, 3 day/week, 90 min low intense physical exercise	Body weight↓, BMI↓	[19]
82 obese women	Uruguay	-	17.20%		12 weeks, 3 day/week, repeated sprint training (130% VO_2_ max 30 s, 55–65% VO_2_ max 3 min)	Absolute and relative maximal oxygen uptake↑, absolute and relative VO_2_ max↑	[196]
35 obesity adolescents	China	12–16 year	14.70%	HiLo	4 weeks, 6 day/week, 2 h/d, Aerobic exercise	Body weight↓, BMI↓, Lean mass↑	[188]
19 overweight or obese females	China	19.30 ± 1.92 year	15%		4 weeks, 3 day/week, 2 h/d	Body weight↓, Fat mass↓, TC↓, adiponectin↓, HDL-C↑	[192]
14 metabolic syndromes	Italy	mean age 55.8 year	16.30%	HiHiLo	2 weeks, 4 day/week, 3 h/d, 55–65% VO_2_ max	TC↓, LDL↓, adiponectin↓, TG↓,	[194]

Abbreviations: FiO_2_: fraction of inspiration O2, DBP: diastolic blood pressure, BMI: body mass index, TG: triglyceride, TC: total cholesterol, HDL: high-density lipoprotein, LDL: low-density lipoprotein, ↑: increase, ↓: decrease. LoHi: living low-training high, IHT: interval hypoxic training, HiLo: living high and training low, HiHiLo: living high-training high-training low.

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
