# Peer review of "Hypoxia as a Double-Edged Sword to Combat Obesity and Comorbidities"

_cells, 2022, doi:10.3390/cells11233735_

Round 1
Reviewer 1 Report
Ruwen Wang and colleagues provide a comprehensive overview of the studies that have been performed to better understand the role of oxygen partial pressure on adipose tissue biology (i.e. adipose tissue inflammation, angiogenesis, remodeling, fibrosis, mitochondrial function) as well as the effects of hypoxia exposure on body weight and cardiometabolic outcomes. Although the topic of this manuscript is of interest and relevant in the field of obesity, I have several major concerns that reduce my enthusiasm for this manuscript.
Major comments:
1) The novelty and added value of the present manuscript is limited in my opinion, since recent reviews on adipose tissue oxygen partial pressure and hypoxia exposure discussing similar concepts have recently been published (Lempesis et al, Acta Physiologica 2020; van Hulten et al, Rev Endocr Metab Disord 2021).
2) It is surprising and problematic that several landmark studies in this field of research are not discussed/cited in this review article. It is stated in paragraph 1.1 that ‘….a large number of studies indicated that oxygen partial pressure was also reduced in adipose tissue, along with the expansion in the tissue mass [9] [10].’ The cited references 9 and 10 are two review papers, published in 2009 and 2014. Since then, several seminal human studies in this field that are not cited in the present manuscript have been published, with conflicting outcomes (i.e. lower, comparable and higher oxygen partial pressure in adipose tissue in obesity) (Pasarica et al, Diabetes, 2009; Goossens et al., Circulation 2011; Hodson et al, Diabetes 2013; Lawler et al, J Clin Endo Metab 2016; Vink et al, Int J Obes 2017; Goossens et al, Diab Obes Metab 2018; Cifarelli et al, J Clin Invest 2020). Moreover, several important studies examining the effects of hypoxia (Lecoultre et al, Diabetes Care 2013; Van Meijel et al, Mol Metab 2021) and hyperoxia (Heilbron et al, Diabetes ; Wilkinson et al, Diab Med 2012; Wilkinson et al, Diving Hyperb Med 2015; Sarabhai et al, Diabetologia 2022) exposure on glucose homeostasis/insulin sensitivity are not cited. Furthermore, several studies that examine the effects of long-term exposure to physiological hypoxia on adipocyte inflammation are not cited/discussed.
3) Many bold statements are made throughout the manuscript, without providing the scientific evidence for it. Just to name a few examples:
- Title paragraph 1.1: ‘Hypoxia in adipose tissue drives the progression of obesity’; this is definitely not proven.
- Paragraph 1.1 and 2.2: ‘…endogenous hypoxic environment in obesity…’; conflicting findings in humans, as outlined above.
- Lines 245-247: Elevation in oxygen consumption during tissue and adipose tissue expansion is essential in catabolism and therefore elicits acute endogenous hypoxia’; not citation is provided to support this statement.
- Lines 302-305: ‘Meanwhile, chronic endogenous hypoxia recruits macrophage into WAT and leukocytes mediate formation of crown-like structures (CLS) [97], which are considered important hallmarks of inflammation.’; a statement that is not supported by evidence/ ref 97.
- Lines 362-363: ‘3.1.2.2. Fibrosis: the feature of white adipose tissue dysfunction induced by hypoxia’: a causal role of oxygen partial pressure in fibrosis has not been proven thus far.
- Lines 418-419: ‘Hypoxia causes mitochondrial dysfunction in white adipose tissue’; what is described in this section, in fact, is the PHYSIOLOGICAL adaptation (not dysfunction of mitochondria!) to lower oxygen availability (i.e. increased reliance on anaerobic glycolysis).
- Figure 1, legend: A causal role for hypoxia in ectopic fat accumulation is not proven.
- Lines 578-580: ‘These findings suggested that the beneficial effects of exogenous hypoxia intervention on obesity and body composition’; please note that the results of several of these studies at high altitude may be explained by important confounders such as lifestyle factors, hypobaric conditions, environmental temperature etc, rather than ‘hypoxia’ per se.
4) Although the authors acknowledge and discuss that the severity of hypoxia as well as the duration of hypoxia exposure may affect study outcomes (sections 5 and 6), this has not been taken into account when interpreting the findings of the in vitro and in vivo studies that are discussed in the previous sections.
5) Figure 2: This figure seems to mix up hypoxia exposure conditions in vitro or in vivo (?). For example, exposure to 15% O2 in the environmental air results in an in situ oxygen level of ~3% in human adipose tissue.
6) The manuscript is difficult to read. The structure as well as writing style could be substantially improved in my opinion. Just some examples: lines 107-111, 115-117, 823-826, 856-858, 861-865, 865-876.
Minor comments:
- Table 1: this Table seems to include studies that investigated the effects of passive hypoxia exposure as well as hypoxia exposure during exercise. I suggest to structure the studies to make this more clear to the reader.
- Certain sections are less relevant (i.e. section ‘2.1 Types of adipose tissue and their functions’
- The authors are encouraged to use people-first language: ‘people with obesity’ instead of ‘obese patients’.
Reviewer 2 Report
This review comprehensively describes the physiological actions of hypoxia in adipose tissue and hypoxic interventions by distinguishing endogenous from exogenous hypoxia. This manuscript is well-written and provides very educative and informative findings.
The reviewer’s concern is only the following point.
Endogenous hypoxia occurs locally in adipose tissue. On the other hand, exogenous hypoxia or hypoxia intervention should systemically impact, which suggests that the effects of exogenous hypoxia on adipose tissue are likely to result from hypoxia-induced changes in other organs or tissues. The reviewer recommends authors to discuss this point in the manuscript.
Reviewer 3 Report
In their review Wang et al. discuss about hypoxia and obesity/adipose tissue. They try to compare the effect of the state of hypoxia that develops during obesity, and which is detrimental to the body (what they call “endogenous hypoxia”) and the observation that exercise in altitude or in low O2 environment tends to be more efficient for weight loss (what the authors call “exogenous hypoxia”). However, as pointed by the authors, (lanes 850-852) “the caveats is that external hypoxia concentrations did not reflect the actual oxygen concentration of adipose tissue”. So since “endogenous hypoxia” and “exogenous hypoxia” are different phenomenon and have no effect in common concerning the adipose tissue and/or obesity this manuscript is actually two different reviews on two different topics.
Lane 187-188 the authors write:
Consistently, BAT activity is significantly and negatively associated with metabolic dysfunction and cardiovascular disease
Since the function of BAT is to burn lipids, it is widely accepted that BAT activity is beneficial for lipid homeostasis.
Lines 269 -271 the authors write
For example, obesity induces secretion of PDGF receptor ß (PDGFR-ß) by inflammatory macrophages in WAT,
Obesity is associated with an increase in PDGF secretion (not PDGF receptor secretion)
In chapter 3.1.2 the authors described the mechanisms by which hypoxia induces insulin resistance. The authors should cite the work of Regazzetti et al. (doi.org/10.2337/db08-0457) that shows that hypoxia creates a state of insulin resistance in adipocytes that is dependent upon HIF transcription factor expression.
Lines 357-360 the authors write
Notably, white adipocytes of obese mice release abundant TGF-ß1, which directly regulates the increase of myofibroblast gene expression (e.g., Acta2 and Tagln) in adipocytes ….
Fibrosis, and the increase in expression of myofibroblastic genes in the adipose tissue of obese animals is mainly (only) observed in myofibroblasts, not in adipocytes.
In chapter 3.1.2.2. “Fibrosis: the feature of white adipose tissue dysfunction “
Since this chapter concern fibrosis the authors must mention and described myofibroblasts of the adipose tissue, which are the cell type responsible for fibrosis.
Lines 511-513 the authors write
Mitochondria in BAT play a key role in BAT to systemically regulate metabolic homeostasis, such as maintaining metabolic homeostasis by regulating BCAA catabolism.
1- What is BCAA?
2- This sentence should be rewritten
Lines 710-729: in this part of the manuscript, which belongs to the “exogenous hypoxia” part, the authors talk about experiments performed on adipocytes in culture treated with 5% O2. The treatment of adipocyte in culture has nothing to do with what the authors call “exogenous hypoxia”, this part belongs to the “endogenous hypoxia” chapter
Lines 742-751: Here again the authors compare mice in low oxygen environment with cells in culture placed under hypoxic conditions. These is unrelated and this part belongs to the “endogenous hypoxia” chapter.
Lines 856-858 the authors write
It is well established that the positive effect of hypoxia in the treatment of chronic diseases such as obesity and metabolic syndrome.
Something is missing in this sentence.
Round 2
Reviewer 1 Report
The authors have substantially improved the manuscript. However, several important comments were not adequately addressed in my opinion. The points below should be addressed to provide the reader with a balanced overview of the scientific evidence, and to assure that results of studies are correctly described and interpreted. Also, I would like to emphasize once more that people-first language should be used throughout the manuscript.
Major comments:
- A major concern I still have with this manuscript is that, despite the fact that the authors correctly acknowledge that findings on adipose tissue partial pressure in humans are conflicting (lines 106-117; hypoxia in the adipose tissue of people with obesity is not ‘established), in several sentences of the Abstract (lines 3-6, lines 10-12), as well as in the Introduction (lines 56-60) and other parts of the manuscript (i.e. lines 72-73; lines 127-129; lines 599-600, lines 893-894etc.), it is still mentioned by the authors that ‘hypoxia is present in obesity’. The same remark applies the titles of section 1.1 and section 2.2, which do not provide a balanced overview of the published literature in my opinion. I suggest that the authors give a more balanced interpretation of the landmark studies in this field of research, and rephrase these sentences such that it becomes clear to the reader that the concept of ‘adipose tissue hypoxia in obesity’ is still not firmly established. Noteworthy, the only human study that has been performed until now to examine weight loss-induced changes in adipose tissue oxygen partial pressure (Vink et al, Int J Obes, 2017) has demonstrated a reduction in oxygen partial pressure in adipose tissue following diet-induced weight loss (which in fact would we in line with higher oxygen partial pressure in obesity). All other studies in this field are cross-sectional in nature. It think that lines 106-117 should be mentioned already in the Introduction.
- Lines 115-117: This statement is not correct. Reading the cited papers, I think the authors mean that ‘lower adipose tissue oxygen consumption in obesity may explain/underlie higher adipose tissue oxygen partial pressure’, and suggest the authors rephrase this sentence accordingly.
- Still, the authors do not report the outcomes of several studies that have examined the effects of longer term exposure to physiological hypoxia on adipocyte inflammation (for example, Famulla S et al, Int J Obes, 2012; Famulla S. et al, Adipocyte, 2012; Vogel M et al, J Clin Endo Metab, 2018). Importantly, these experiment may better reflect the human in vivo situation compared to other in vitro studies that have exposed (pre)adipocytes to (severe) hypoxia in an acute setting.
- Section 6.1: I suggest that the authors also cite the recent systematic review by Van Hulten et al, Rev Endocr Metab Disord 2021, in which the results of studies that examined the effects of hypoxic exercise on glucose metabolism are reported and discussed.
- Section 6.2:
· Reference 253 (Lecoultre et al) was a within-group comparison (i.e. no control group was included), which may be important to mention here.
· Results related to reference 252 (Van Meijel et al) should be reported more accurately. That study found no significant changes in plasma glucose and insulin concentrations, and no insulin sensitivity in adipose tissue, muscle and liver, but demonstrated that hypoxia exposure increased insulin-independent glucose uptake in human primary myotubes through AMPK.
· References 254-256: could the difference be explained by the HYPERBARIC condition (i.e. references 252-253 used exposure to normobaric hypoxia)?
- Still, the authors do not consistently use people-first language (i.e. ‘people with obesity’ instead of ‘obese subjects’), for example in line 109.
- Line 570: As mentioned previously, there is no evidence that ‘hypoxia may aggravate obesity’.
- Line 16: brought by hypoxia exercise’ should read ‘brought by hypoxic exercise’
Reviewer 3 Report
The authors have addressed all my comments. I think that this review will be of interest for researchers working on the field.
There are however some minor problems in the bibliography:
line 1011: the name of the authors are missing
Line 1034 reads “20. . (!!! INVALID CITATION !!! [20]). »
Line 1628 : the name of the authors are missing
